# A Reliability System Evaluation Model of NoC Communication with Crosstalk Analysis from Backend to Frontend

**DOI:** 10.3390/mi14020469

**Published:** 2023-02-17

**Authors:** Xiaodong Weng, Xiaoling Lin, Yi Liu, Changqing Xu, Linjun Zhan, Shunyao Wang, Dongdong Chen, Yintang Yang

**Affiliations:** 1School of Microelectronics, Xidian University, Xi’an 710126, China; 2Technology on Reliability Physics and Application Technology of Electronic Component Laboratory, China Electronic Product Reliability and Environmental Testing Research Institute, Guangzhou 510610, China; 3Guangzhou Institute of Technology, Xidian University, Guangzhou 510555, China

**Keywords:** network on chip, interconnects, crosstalk, hybrid automatic repeat request

## Abstract

Network on chip (NoC) is the main solution to the communication bandwidth of a multi-processor system on chip (MPSoC). NoC also brings more route requirements and is highly prone to errors caused by crosstalk. Crosstalk has become a major design problem in deep-submicron NoC communication design. Hence, a crosstalk error model and corresponding reliable system with error correction code (ECC) are required to make NoC communication reliable. In this paper, a reliability system evaluation model (RSE) of NoC communication with analysis from backend to frontend has been proposed. In the backend, a crosstalk error rate model (CER) is established with a three-wire RLC coupling model and timing constraints. The CER is used to establish functional relations between interconnect spacing, length and signal frequency, and test system reliability. In the frontend, a reliability system performance model (RSP) is established with a CER, reliability method cost and bandwidth. The RSE summarizes the frontend and backend model. In order to verify the RSE model, we propose a reliability system with a hybrid automatic repeat request technique (RSHARQ). Simulation demonstrates that the CER model is close to real circuit design. Through the CER and RSP model, the performance of RSHARQ could be simulated.

## 1. Introduction

Network on chip (NoC) has emerged as a promising solution for multicore system on chip communication design [1,2], which overcomes the complexity of wire delay and flexibility in chip architectures [3,4,5,6]. NoC is a packet-based, on-chip communication switching network designed for communication among the intellectual property (IP) cores of SoC systems [7]. NoCs use packets to exchange data between processing elements (PEs) via network fabric that consists of resource network interfaces (RNI), routers and interconnecting links. Among these, interconnection links are the most affected by crosstalk [8].

Crosstalk is a type of noise which is introduced by an unwanted coupling between a node and its neighboring wire or between two neighboring wires [9]. As technology scales down, the dimensions of interconnects also scale down. This reduces the spacing among interconnects [10]. With the high signal frequency and small spacing between interconnects, the coupling between the lines has become well known [11,12]. Coupling between signal lines can cause logic failures and timing degradation in digital systems [13]. Crosstalk has become the most critical concern in modern sub-10 nm integrated circuits [14]. This coupling causes problems that can be addressed by various methods like shielding [15,16,17,18,19,20], inserting repeaters [21,22,23,24,25] or buffers [26,27,28,29,30,31], duplication [32] and crosstalk avoidance code (CAC) [33,34,35,36]. Insertion of buffers and shielding has more area cost when compared to duplication or CAC [37].

Various error correction code (ECC) schemes have been proposed to protect communication between IP cores [38,39,40,41]. In [38,39], the authors focus on weighing the performance and design cost of ECC. In [40,41], which address improving the transient fault tolerance, have shown that implementing complex ECC schemes incurs high area overhead and high energy dissipation and may adversely affect the performance of the NoC. They use assumed crosstalk error injection to verify their schemes. The error rate caused by crosstalk is relevant to the spacing between interconnects and the frequency of NoC. Bandwidth is mainly determined by spacing between interconnects and working frequency. Hence, a crosstalk error model and corresponding reliable system with ECC are required to make NoC communication reliable.

In this paper, a reliability system evaluation model of NoC communication with analysis from backend to frontend has been presented (BFRS). In the backend, with a three-wire RLC coupling model, an error rate model considering timing constraints (SSTC) is proposed to establish functional relations between interconnect spacing, length and signal frequency. With different nodes’ dimensions, the SSTC calculates out error rates modeled with the physical size of interconnects and the working environment, which can be used to test the system’s reliability. In the frontend, effective bandwidth can be modeled with route spacing, working frequency, encode cost on code length and automatic repeat-request (ARQ) cost with error injection rates. The reliability system performance model (RSP) can be modeled with bandwidth and effective bandwidth. BFRS summarizes the frontend to backend model. In order to verify BFRS model, we propose a reliability system with a hybrid automatic repeat request technique (RSHARQ). RSHARQ uses Hamming code as ECC and a cyclic redundancy check (CRC) with the proposed code transmission scheme as ARQ. Finally, simulation results are shown.

## 2. The Methods of Establishing RSP Model and RSHARQ

### 2.1. Three-Wire Coupling Interconnect Model

Referring to [10,42], Section 2.1 establishes the RLC coupling model of interconnects, as Figure 1 and Figure 2 show.

Figure 1 is the cross-sectional view of the coupling interconnect model. Figure 1a shows the equivalent capacitance model of coupling interconnects. $C_{g}$ is the coupling capacitance between an interconnect and the grounding metal layers. $C_{c}$ is the coupling capacitance of adjacent interconnects. Figure 1b shows the dimensions of an interconnect.

Figure 2 shows the distributed RLC spice model of an interconnect, where the interconnect is equivalently divided into n segments. $V_{\mathrm{in}}$ is the driving voltage source, $R_{\mathrm{th}}$ is the driving resistance, and $C_{f}$ is the equivalent load of the interconnect.

All parameters mentioned above can be calculated from the interconnect dimensions shown in Figure 1b, the material parameters, the working status parameters (called timing constraint parameters as well) and the formulas listed below.

Dimension parameters of an interconnect include: the length l, width w, thickness t, spacing s and distance h. Material parameters include: the insulating layer relative dielectric constant $\varepsilon_{r}$ and interconnect resistivity ρ. Working status parameters include: the rising time $t_{r}$ and falling time $t_{f}$.

Referring to [42,43,44,45], formulas are shown below:(1)$$R=\frac{\rho \;\cdot \;l}{w\;\cdot \;t}$$
(2)$$L=\frac{\mu_{0}\;\cdot \;l}{2\pi }\left[\ln\left(\frac{2l}{w+t}\right)+\frac{1}{2}+\frac{0.22\left(w+t\right)}{l}\right]$$
(3)$$M=\frac{\mu_{0}\;\cdot \;l}{2\pi }\left[\ln\left(\frac{2l}{s+w}\right)-1+\frac{s+w}{l}\right]$$
(4)$$k_{M}=\frac{M}{L}$$
(5)$$C_{g}=\varepsilon_{0}\varepsilon_{r}\left[\frac{w}{h}+2.04{\left(\frac{s}{s+0.54h}\right)}^{1.77}\cdot \;{\left(\frac{t}{t+4.53h}\right)}^{0.07}\right]$$
(6)$$C_{c}=\varepsilon_{0}\varepsilon_{r}\left[1.41\frac{t}{s}e^{-\frac{4s}{s+8.01h}}+2.37{\left(\frac{w}{w+0.31s}\right)}^{0.28}\cdot \;{\left(\frac{h}{h+8.96s}\right)}^{0.76}\cdot \;e^{-\frac{2s}{s+6h}}\right]$$
(7)$$n\geq 10\left(\frac{l}{V\;\cdot \;t_{r}}\right)$$
where R is the resistance of the whole interconnect, L is the self-inductance of interconnects, M is the mutual inductance between interconnects, $\mu_{0}$ is the vacuum permeability, $k_{M}$ is the mutual inductance coefficient, $\varepsilon_{0}$ is the vacuum dielectric constant, V is the propagation velocity of electromagnetic wave in interconnects, and n is the number of the three-wire distributed RLC spice model’s interconnect segments.

### 2.2. An Error Rate Model with RLC Wire Coupling Model and Timing Constraints (SSTC)

With the three-wire RLC coupling model of interconnects and different bestirring sources on the three wires, wire delays could be simulated out with spice. In this section, the measurement method of crosstalk errors and error rates are presented.

In digital circuit design, we use the timing constant parameters input delay and output delay to describe the delay caused by signals passing through wires and ports. Input delay means delay caused by signals passing from outside to inside through ports. Output delay means delay caused by signals passing from inside to outside through ports. Generally, we set the input delay and output delay as 60% of the timing cycle length. The delay of signals passing through ports is set as 20% of the cycle length, so the wire delay is 40% of the cycle length.

With the bestirring source frequency, the timing cycle length can be obtained. If the signal wire transmission delay is greater than 40% of the timing cycle length through spice simulation, coupling crosstalk will cause bit errors on the signal line.

With simulation results of all possible bestirring sources, we can build the SSTC model.

Through the SSTC model, an interconnect maximum operating frequency is shown in Equation (8):(8)$$f_{m}=\frac{0.4}{t_{\mathrm{tdm}}}$$
where $t_{\mathrm{tdm}}$ is the delay of transmission through wires. $f_{m}$ is the maximum working frequency.

### 2.3. A Reliability System Performance Model (RSP)

Bit errors can be corrected or detected by a reliability system. ECC is used to correct n-bit errors and detect n + 1-bit errors; ARQ is used to detect n-bit errors without the ability to correct errors, where n is the length of the error bits. If errors detected could not be corrected, the system will send data again. HARQ is a scheme combining ECC and ARQ. Considering ECC and ARQ integration, a reliability system performance model with relationship evaluation between error rates, reliability method cost and bandwidth has been presented. Details of the RSP model are shown below.

Most parameters mentioned in Section 2.1 are decided by chip fabrication technology. The designer can only decide the spacing and length of interconnects. For a given length of an interconnect, crosstalk error rates e can be expressed as function g, as Equation (9) shows:(9)$$e=g\left(s,f_{m}\right)$$

In digital circuit design, the spacing between interconnects determines the maximum amount of route lines and the max bandwidth B. Equation (9) can be modified into Equation (10):(10)$$e=g\left(\frac{1}{B},f_{m}\right)$$

Assuming that a reliability system with HARQ could correct a bit errors and detect b bit errors in a code of length n bits, where b is larger than a, the possibilities of single transmission success $P_{a}$ and system accurate judgment $P_{b}$ can be calculated with Equations (10) and (11).
(11)$$P_{a}=\overset{a}{\underset{m=0}{\sum }}\frac{n!}{m!\left(n-m\right)!}\;\cdot {\;e}^{m}{\left(1-e\right)}^{n-m}$$
(12)$$P_{b}=\overset{b}{\underset{m=0}{\sum }}\frac{n!}{m!\left(n-m\right)!}\;\cdot {\;e}^{m}{\left(1-e\right)}^{n-m}$$

The possibility of resending data in single transmission is P:(13)$$P=P_{b}-P_{a}$$

The possibility of reliability scheme failure is $P_{f}$:(14)$$P_{f}=1-P_{b}$$

The expected number of one-packet transmission times is $N_{t}$:(15)$$N_{t}=1\;\cdot \;P+2\;\cdot \;P\left(1-P\right)+3\;\cdot \;P{\left(1-P\right)}^{2}+\cdots +l\;\cdot \;P{\left(1-P\right)}^{l-1}+\cdots =\frac{1}{P}$$

The effective throughput of the HARQ reliability system is $\eta_{\mathrm{SR}}$:(16)$$\eta_{\mathrm{SR}}=\frac{1}{N_{t}}\left(\frac{k}{n}\right)=\left(\frac{k}{n}\right)P$$
where k is the original code length without ECC encoding.

As a result, the effective bandwidth $B_{w}$ can be calculated as follows:(17)$$B_{w}={\;\eta }_{\mathrm{SR}}\;\cdot \;B$$

With the formula mentioned above, the RSP model has been built to explain the relationship between error rates, reliability method cost and bandwidth.

Summarizing the SSTC model and RSP model, a reliability system evaluation model (BFRS) is established from backend to frontend.

### 2.4. A Reliability System with HARQ and CRC Technique (RSHARQ)

In order to verify the BFRS model, we develop a reliability system with HARQ and the CRC technique (RSHARQ). The system flowchart is shown in Figure 3. The details of RSHARQ are shown below.

Define one packet with 4 flits, and each flit has 5 bits. In 20 bits, transmission information occupies 16 bits, while the rest is for urgent words or CRC words. In order to improve the system reliability, the lines for CRC words or urgent words should not be influenced by crosstalk. Each flit has 4 information bits and a special bit.

In the first operation cycle, the system sends 16 bits information and 4 bits CRC words. If the CRC check is right, the transmission is successful. If not, the system sends a packet with each flit’s Hamming word, which has a length of 12 bits. If the ECC detects a multi-error, the system will execute ARQ for each flit. If the CRC check is still false with ECC correcting, the system will execute ARQ.

## 3. Results

The simulation was carried out with the University of California, Berkeley’s Predictive Technology Model (PTM) 65 nm nodes and a TSMC 28 nm High-Performance Computing (HPC) library. Table 1 shows values of a PTM 65 nm interconnect. Table 2 shows values of a TSMC 28 nm interconnect. Notably, 65 nm node values are used to verify the processes of the SSTC and RSP model. Due to the completed library files of 28 nm nodes, we compare the SSTC model result with parameters extracted from the backend.

The simulation adopts the Cadence process to work out. INNOVUS and QRC are used to simulate the backend mentioned in the SSTC model. SPECTRE is used for Hspice simulation. Additionally, MATLAB is used to calculate the parameters mentioned in the models and simulate the frontend mentioned in the RSP model and RSHARQ.

Firstly, 65 nm node simulation is presented. The driven voltage is set to 1.2 V, the equivalent drive resistance is set to 156 Ω, and the equivalent load capacitance is set to 64 fF. The wire length is set as 500 µm.

Directional arrows are used to indicate the bestirring source. For example, “-“ means the signal of the wire is not changing, “↑” means the signal of the wire is changing from low voltage to high voltage, and “↓” means the signal of the wire is changing from high voltage to low voltage. We use directional arrow aggregates to express the bestirring source on the three-wire coupling model, such as (-, ↑, ↓).

Table 3 shows the three-wire coupling model spice simulation results on a 65 nm node.

From Table 3, the delay can be divided into (120, 110, 95, 80, 74, -). For convenience of description, the delay around 120 ps called G6 delay, which occurs in the middle line with two different signals changing direction in the side lines, as (↑, ↓, ↑). Similarly, the delay around 110 ps is called G5 delay, the delay around 95 ps is called G4 delay, the delay around 80 ps is called G3 delay, the delay around 74 ps is called G2 delay, and the delay around 0 is called G1 delay. With Formula (8), the G6 frequency, denoted as G6. Freq, is 0.4/120 ps = 3.3 GHz. Similarly, G5. Freq is 3.6 GHz, and G4. Freq is 4.2 GHz.

If the working frequency is in the range of (G5. Freq, G6. Freq) and not equal to G6. Freq, all situations of G6 delay lead to error. The system error is denoted as a G6 crosstalk error. Similarly, a system error with a working frequency in the range (G5. Freq, G4. Freq) and not equal to G5. Freq is denoted as a G5 crosstalk error. As shown in Table 3, the G6 crosstalk error rate is 0.97%, and the G5 crosstalk error rate is 5.21%.

In Table 4, we simulate the relationship between bandwidth and wire dimensions (spacing and length), which can be modified by the designer. A B1 rate means bandwidth improvement rates between bandwidth, with the wire spacing changing and the original bandwidth with G6. Freq. A B2 rate means bandwidth improvement rates between bandwidth with G5. Freq and bandwidth with G6. Freq under the same spacing.

Table 4 shows that both the B1 rate and B2 rates increase with the decrease of wire spacing. However, with spacing decreasing, the maximum frequency is lower as well.

Then, we give simulation results on 28 nm nodes. The driven voltage is set to 0.9 V, the equivalent drive resistance is set to 156 Ω, and the equivalent load capacitance is set to 4.5 pF on 28 nm nodes. The wire length is set to 800 µm. We build a real three-wire model in the Cadence process, as Figure 4 shows.

Figure 4c is the layout and top view of the wire model design. In Figure 4c, INV11, INV12, INV13, INV21, INV22 and INV23 are library standard cells, which are displayed as gray areas located in the space between VSS and VDD.

Table 5 shows the three-wire coupling model spice simulation results and real model extraction results on 28 nm nodes. The spice model results are recoded as “-S”, while the real model results are recoded as “-R”.

Table 5 shows that the delay with spice simulation is close to the delay with INNOVUS extraction, which means the three-wire coupling spice model is close to the real circuit model.

Additionally, the frequency, error rate and bandwidth are analyzed.

From Table 5, the delay can be divided into (874, 701, 494, 334, 236, -). For convenience of description, the delay around 874 ps is called G6 delay, which occurs in the middle line with two different signals changing direction in the side lines, as (↑, ↓, ↑). Similarly, the delay around 701 ps is called G5 delay, the delay around 494 ps is called G4 delay, the delay around 334 ps is called G3 delay, the delay around 236 ps is called G2 delay, and the delay around 0 is called G1 delay. With Formula (8), the G6 frequency, denoted as G6. Freq, is 0.457 GHz. Similarly, G5. Freq is 0.570 GHz.

If the working frequency is in the range of (G5. Freq, G6. Freq) and not equal to G6. Freq, all situations of G6 delay lead to errors. The system error is denoted as G6 a crosstalk error. Similarly, a system error with the working frequency in the range (G5. Freq, G4. Freq) and not equal to G5. Freq is denoted as a G5 crosstalk error. As shown in Table 3, the G6 crosstalk error rate is 0.97%, and the G5 crosstalk error rate is 5.21%.

In Table 6, we simulate the relationship between bandwidth and wire dimensions (spacing and length), which can be modified by the designer. A B1 rate means bandwidth improvement rates between the bandwidth, with the wire spacing changing and the original bandwidth with G6. Freq. B2 rate means bandwidth improvement rates between the bandwidth with G5. Freq and the bandwidth with G6. Freq under the same spacing.

Table 6 shows that both the B1 rate and B2 rate increase with the decrease of wire spacing. However, with spacing decreasing, the maximum frequency is lower as well.

The simulation results for the 28 nm node are similar to those for the 65 nm node.

Generally, the reliability system has the possibility to break down when the occurring error is out of the design range. The breakdown phenomenon is called collision. In Table 7, (20,16) the CRC collision rate is simulated with different injection bit error rates.

Table 7 shows that under G6 crosstalk error, the CRC has a 1.9‱ chance to break down; under a G5 crosstalk error, the CRC has a 50.2‱ chance to break down.

Table 8 shows average sending times and collision times of RSHARQ with different error rates under 100,000 cycles of simulation.

Under the G6 crosstalk error rate, RSHARQ has a 17.4% cost to ensure data reliability. Under the G5 crosstalk error rate, RSHARQ has a 80.3% cost to ensure data reliability. Assuming that RSHARQ is designed on 65 nm nodes, according to the system frequency required, the spacing of lines could be chosen and the effective bandwidth of system could be calculated out before the real design is implemented.

## 4. Discussion

In Section 3, Table 4 evaluates the bandwidth change with the B1 rate and B2 rate. The B1 rate represents the impact of interconnect spacing changes on bandwidth. The B2 rate represents the bandwidth improvement ratio before and after the hardening design. Through the simulation, the B1 rate increases with the decrease of spacing, which means that although the frequency decreases with the decrease of interconnect spacing, the bandwidth of interconnects increases with the decrease of spacing. The B2 rate decreases with the decrease of spacing, which means that the bandwidth improvement ratio of the hardening design decreases. The reliability design guarantees the accuracy of signal transmission and allows the decrease of the interconnection distance while maintaining the original frequency. The bandwidth is increased by tolerating a part of the crosstalk error with the reliability design. With the decrease of the interconnection spacing, the improvement of this method become lesser.

For NoC design, we hope that each IP has enough bandwidth allocation. In fact, the area of a chip is limited, which leads to the limitation of the wiring area and bandwidth for each IP. The current digital integrated circuit design, especially regarding router design with large traffic, often adopts a lower frequency and smaller interconnection spacing in order to pursue greater bandwidth. However, the premise of adopting this method in the stage of layout is to change the NoC design from an isomorphic design to a heterogeneous design, which increases the complexity of router design.

In this paper, a BFRS is proposed to formulate the relationship between bandwidth and interconnect parameters, which determines the chip area with the IP area. By using the proposed model in the analysis stage of the integrated circuit library, an evaluation model for length, spacing, working frequency and bandwidth could be established. The length and spacing of the interconnects between each route are determined by combining the area size and bandwidth demand analysis of the IP selected.

## 5. Conclusions

A reliability system evaluation model of NoC communication with analysis from backend to frontend (BFRS) has successfully been validated against the well-established model. With the model proposed and a simulation of a PTM 65 nm library, we establish the crosstalk error rate model and the relationship between interconnect spacing, length, working experience and bandwidth. The B1 rate and B2 rate are proposed to evaluate the change rate of the bandwidth. In order to verify the accuracy of proposed model, the interconnect parameters are simulated with the proposed model and simulated with a real circuit in a TSMC 28 nm library. The simulation results show that the SSTC spice model is close to the real circuit model.

In the stage of analysis of the library, with the BFRS model, we could obtain a preliminary judgment on chip area, bandwidth, frequency and reliability with the selected IP’s information. The BFRS model can help designers to evaluate the reliability system with real error rates using selected technology nodes, rather than assuming the error injection model.

Meanwhile, in order to evaluate the crosstalk influence in a real circuit, parasitic extraction and a circuit spice model are required. With the development of Golden Spice+GPU and Fast Spice, spice simulation analysis can be carried out on key modules of digital circuits. The simulation speed cannot meet the requirements of digital integrated circuit system verification, and the reliability of the system cannot be fully verified. With model proposed and delay replacement, in the early design period, the crosstalk delay influence can be evaluated with transfer simulation.

## 6. Patents

Patents “A highly Reliable Communication System for Network on Chip of Multicore Processor System” (No. CN202210261693.4) and “A Design Performance Evaluation Method and System of NoC Communication Architecture” (No. CN202210253790.9) are resulting from the work reported in this manuscript.

## Figures and Tables

**Figure 1 micromachines-14-00469-f001:**
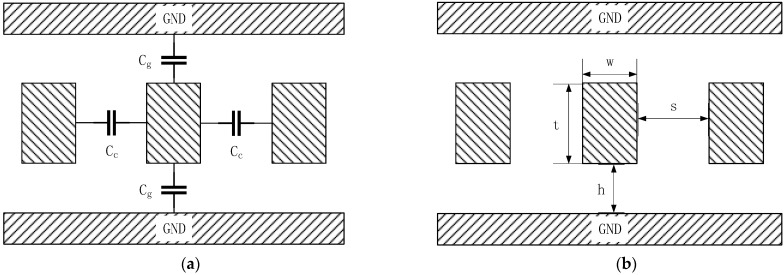
A coupling interconnect model: (**a**) is the equivalent capacitance model; (**b**) is the dimensions of an interconnect.

**Figure 2 micromachines-14-00469-f002:**
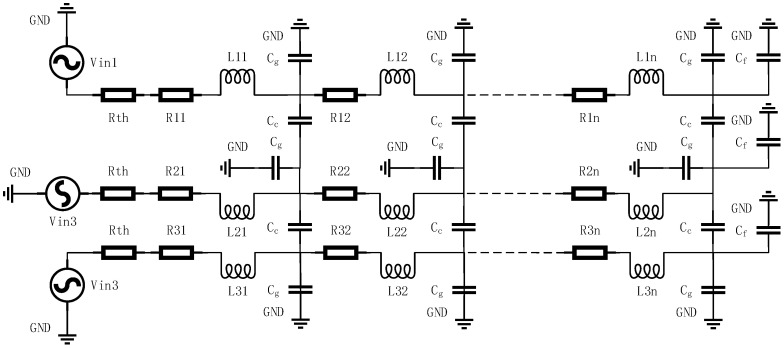
Distributed three-wire RLC spice model.

**Figure 3 micromachines-14-00469-f003:**
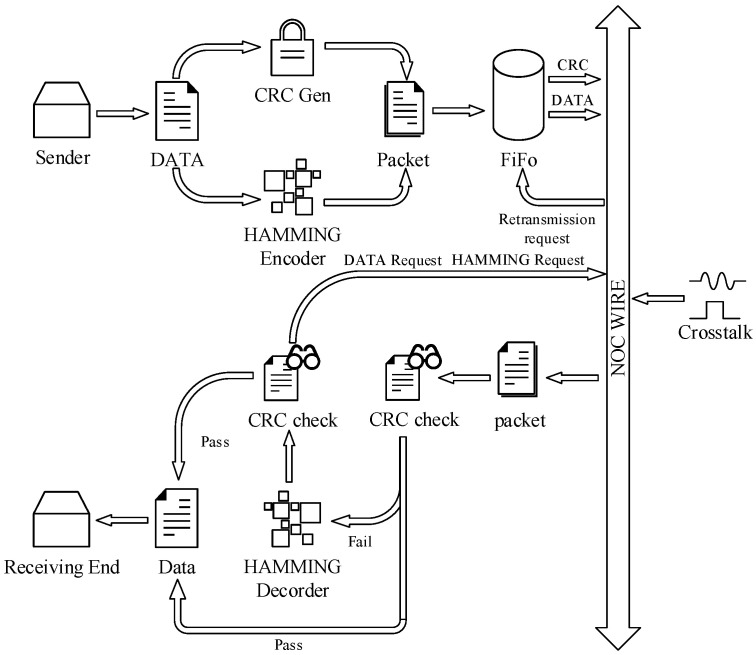
The flowchart of RSHARQ.

**Figure 4 micromachines-14-00469-f004:**
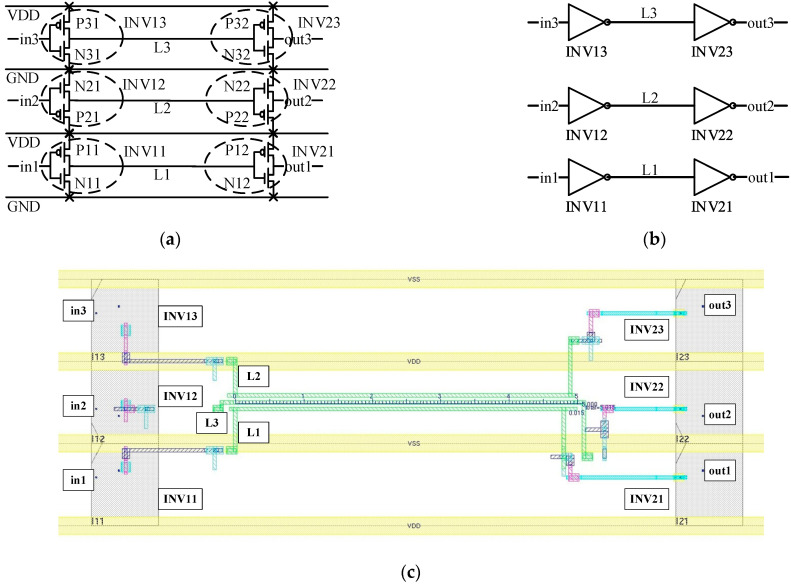
Real three-wire model realization with INNOVUS: (**a**) is the circuit diagram; (**b**) is the standard cell logic diagram; (**c**) is the layout graph.

**Table 1 micromachines-14-00469-t001:** PTM 65 nm interconnect values.

Dimensions/µm		Material
w = 0.14	s = 0.14	$\rho =2.2\times 10^{-8}\Omega /m$
h = 0.20	t = 0.35	$\varepsilon_{r}=2.2$

**Table 2 micromachines-14-00469-t002:** TSMC 28 nm interconnect values.

Dimensions/µm		Material
w = 0.05	s = 0.05	$\rho =4.02\times 10^{-8}\Omega /m$
h = 0.085	t = 0.09	$\varepsilon_{r}{}^{1}=2.63/3.55$

^1^ In 28 nm library, the $\varepsilon_{\mathrm{rc}}$ of $C_{c}$ is 2.63 and the $\varepsilon_{\mathrm{rg}}$ of $C_{g}$ is 3.55.

**Table 3 micromachines-14-00469-t003:** Wire delay of 65 nm three-wire coupling model.

Signal	Left Wire/ps	Mid Wire/ps	Right Wire/ps
(-, -, -)	-	-	-
(-, -, ↑)	-	-	76.17
(-, -, ↓)	-	-	76.17
(-, ↑, -)	-	93.7	-
(-, ↑, ↑)	-	72.95	56.97
(-, ↑, ↓)	-	110.2	95.38
(-, ↓, -)	-	93.7	-
(-, ↓, ↑)	-	110.2	95.38
(-, ↓, ↓)	-	72.95	56.97
(↑, -, -)	76.17	-	-
(↑, -, ↑)	73.43	-	73.43
(↑, -, ↓)	80.23	-	80.23
(↑, ↑, -)	56.97	72.95	-
(↑, ↑, ↑)	53.91	53.56	53.91
(↑, ↑, ↓)	60.52	93	98.63
(↑, ↓, -)	95.38	110.2	-
(↑, ↓, ↑)	94.04	122.4	94.04
(↑, ↓, ↓)	98.63	93	60.52
(↓, -, -)	76.17	-	-
(↓, -, ↑)	80.23	-	80.23
(↓, -, ↓)	73.43	-	73.43
(↓, ↑, -)	95.38	110.2	-
(↓, ↑, ↑)	98.63	93	60.52
(↓, ↑, ↓)	94.04	122.4	94.04
(↓, ↓, -)	56.97	72.95	-
(↓, ↓, ↑)	60.52	93	98.63
(↓, ↓, ↓)	53.91	53.56	53.91
(-, ↓, -)	-	93.7	-
(-, ↓, ↑)	-	110.2	95.38
(-, ↓, ↓)	-	72.95	56.97
(↑, -, -)	76.17	-	-
(↑, -, ↑)	73.43	-	73.43
(↑, -, ↓)	80.23	-	80.23

**Table 4 micromachines-14-00469-t004:** SSTC SPICE simulation and bandwidth change on 65 nm node.

	Original	0.9 Spacing	0.8 Spacing	0.7 Spacing	0.6 Spacing	0.9 Length	0.8 Length	0.7 Length
Delay	124.6 ps	129.4 ps	137.1 ps	146.0 ps	159.1 ps	112.0 ps	101.0 ps	89.9 ps
G6 Delay	103.0 ps	107.0 ps	113.0 ps	123.0 ps	135.0 ps	93.8 ps	84.4 ps	75.3 ps
G6. Freq	3.21 GHz	3.09 GHz	2.92 GHz	2.75 GHz	2.51 GHz	3.58 GHz	3.96 GHz	4.45 GHz
G5. Freq	3.90 GHz	3.74 GHz	3.53 GHz	3.27 GHz	2.96 GHz	4.27 GHz	4.74 GHz	5.31 GHz
B1 rate	0.000	0.069	0.136	0.223	0.303	0.070	0.136	0.223
B2 rate	0.216	0.209	0.209	0.189	0.179	0.190	0.197	0.193

**Table 5 micromachines-14-00469-t005:** Wire delay spice and extraction results of SSTC on 28 nm nodes.

	Left Wire-S/ps	Mid Wire-S/ps	Right Wire-S/ps	Left Wire-R/ps	Mid Wire-R/ps	Right Wire-R/ps
(↓, ↑, -)	553	679	-	531	682	-
(-, ↑, ↑)	-	679	553	-	682	531
(↑, ↓, -)	519	736	-	514	701	-
(-, ↓, ↑)	-	373	519	-	701	514
(↑, ↓, ↑)	492	909	492	487	874	496
(↓, ↑, ↓)	535	854	535	513	857	522
(↑, ↓, ↓)	529	529	304	524	494	282
(↓, ↓, ↑)	304	529	529	282	494	524
(↓, ↑, ↑)	566	489	282	544	492	277
(↑, ↑,↓)	282	489	566	277	492	544
(-, -, ↓)	-	-	386	-	-	364
(↓, -, -)	386	-	-	364	-	-
(-, -, ↑)	-	-	359	-	-	354
(↑, -, -)	359	-	-	354	-	-
( -, ↑, -)	-	489	-	-	492	-
(-, ↓, -)	-	531	-	-	496	-
(-, -, -)	-	-	-	-	-	-
(↓, -, ↓)	362	-	362	340	-	340
(↑, -, ↑)	331	-	331	326	-	326
(↓, -, ↑)	412	-	382	390	-	377
(↑, -, ↓)	382	-	412	377	-	390
(-, ↑, ↑)		331	257	-	334	252
(↑, ↑, -)	257	331	-	252	334	-
(-, ↓, ↓)	-	359	271	-	324	249
(↓, ↓, -)	271	359	-	249	324	-
(↑, ↑, ↑)	233	233	233	228	236	228
(↓, ↓, ↓)	256	256	256	234	221	234

**Table 6 micromachines-14-00469-t006:** SSTC SPICE simulation and bandwidth change on 28 nm node.

	Original	0.9 Spacing	0.8 Spacing	0.7 Spacing	0.6 Spacing	0.9 Length	0.8 Length	0.7 Length
Delay	874	946	1036	1156	1316	709	560	430
G6 Delay	701	752	821	909	1036	566	447	338
G6. Freq	0.457	0.423	0.396	0.346	0.304	0.564	0.714	0.930
G5. Freq	0.570	0.532	0.487	0.440	0.386	0.707	0.895	1.183
B1 rate	0	0.026	0.054	0.080	0.107	0.233	0.561	1.032
B2 rate	0.247	0.258	0.262	0.270	0.272	0.252	0.253	0.272

**Table 7 micromachines-14-00469-t007:** Possibility of (20,16) CRC collision.

**Bit Error Rate/%**	0.5	1.04	2	3	4	4.5	5	5.21
**Collision Possibility/‱**	0.1	1.9	6.5	14	28.2	33.4	46.5	50.2

**Table 8 micromachines-14-00469-t008:** Average sending times of different error rates.

**Bit Error Rate/%**	0.1	0.3	0.5	1.04	2	3	4	4.5
**Average Sending Times**	1.0178	1.049	1.086	1.174	1.341	1.511	1.704	1.803
**Effective Throughput**	0.983	0.953	0.921	0.852	0.746	0.662	0.587	0.555

## Data Availability

Data are available from the authors upon reasonable request.

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
