# Peer review of "A Reliability System Evaluation Model of NoC Communication with Crosstalk Analysis from Backend to Frontend"

_micromachines, 2023, doi:10.3390/mi14020469_

Round 1

Reviewer 1 Report

The paper deals with a reliability system evaluation model of NoC communication 2 with crosstalk analysis from backend to frontend. The Introduction provides the inevitable analysis of the described topic. The paper is well-written, and its organization has a high standard. The paper's main idea is accurate. Even while I consider the paper to be of high quality, I do have a few comments:

1, The only basis for the results presented in the paper are simulations. Why the authors did not present the results of real measurements, which can demonstrate the accuracy of the simulation results?

2, 23 references are insufficient for this type of paper.

3, Figure 5 is not readable.

4, The Discussion section of the paper is too short.

5, The Conclusion is not present.

Reviewer 2 Report

I have the following comments on the paper titled "A Reliability System Evaluation Model of NoC Communication with Crosstalk Analysis from Backend to Frontend."

  • Readers would like to have a better understanding of the state-of-the-art results, what are the shortcomings of the state-of-the-art that the proposed techniques overcome, what is the novelty of the proposed technique, the comparison of the results with state-of-the-art results, etc. However, such aspects are not clearly presented by the authors.
  • Section 2.1: 
    • Is the "Three-wire coupling interconnets model" universally accepted?
    • Are all three wires running from the driver to the driven node?
    • The model resembles some sort of transmission line. Can't we directly borrow results from the transmission line theory? Or have authors done the same without adequate citations?
    • Can't figures 2 and 3 be clubbed?
    • Equations (1) - (17): are these the contributions of the authors? If not, proper references are required.
  • Section 2.2: 
    • "Generally, we set input delay and output delay as 60% timing cycle length. The delay of signal passing ports is set as 20% cycle length, so wire delay is 40% cycle length" Is this a universally accepted practice? If not, what is the basis of this?
  • Section 2.3:
    • "Generally, ECC is used to correct n bits errors and only detect n+1 bits error" What is n here? Is this a global rule?
    • Equation 10: Isn't the maximum operating frequency and bandwidth related? Or is there a reason for keeping B and fm as independent variables in this equation?
    • It reads, "The expected number of one packet transmission times is T_SR," after equation (14). Is T_SR a measure of time or a number? What's the significance of SR?
    • Figure 4: Fail is shown as Fall.
  • Section 3: 
    • It is mentioned that "PTM 65nm nodes and TSMC 28nm hcp 1ibrary" (There are typos here "65nm nodes" to "65nm node", "1ibrary" to "library". Why do the authors want to include the results of two different nodes? Since 65nm is PTM and 28nm is foundry models, can't we have the results for just the 28nm node?
    • Tables 1 and 2: What's the basis for choosing these values?
    • "In 28nm library, the εr of Cc is 2.63 and the εr of Cg is 3.55." One would expect to see the same εr since the material is the same. How do the authors justify such a significant difference?
    • Page 7: "equivalent drive resistance is set to 156 Ω, and equivalent load capacitance is set to 64fF." How did the authors choose these values?
    • Page 8: Is the G6 frequency related to G6 delay? How does one calculate the maximum frequency of operation from the delay?
    • Page 8: Ghz to GHz
    • The load capacitance for the 65nm node is set to 64fF, and for 28nm, it is set to 4.5pF. Why is there a huge difference?
    • Figure 5: It is difficult to relate this to figures 1 and 2. Are there ground layers above and below the three pink metals? Won't the dark blue vdd and vss affect the capacitance of the pink paths?
    • Table 5: Why doesn't it have all the transitions as in table 3?
  • The language and style need thorough checks. Some issues in the abstract are listed below.
    • NoC (Network on Chip) to Network on chip (NoC) - Avoid random capitalization
    • Not easy to understand the abbreviation-full form relationships of BFRS, SSTC, RSP, etc.
    • Some sentences are not grammatically correct. E.g. "Crosstalk has become a major challenge in deep-submicron on-chip communication, especially NoC"

Round 2

Reviewer 1 Report

The paper has been much improved and I recommend to accept the paper in the current form.

Author Response

Thank you for your comments.  The description of  method is further revised.

Reviewer 2 Report

The authors have addressed most of my comments. I still have some minor remarks, listed below.

  • There was a typo in my earlier comments. The original was supposed to be, "Can't figures 1 and 3 be clubbed?"
  • "Figure 5: It is difficult to relate this to figures 1 and 2." It would have been better if the authors mentioned these as the top view, cross-sectional view, etc.
  • I am still trying to understand why PTM 65nm results are needed if 28nm results are there.
  • Aren't there any other wire-delay prediction models available? It would have been interesting if Table 5 had a comparison between the proposed technique, the existing technique, and the extracted results. It would also be interesting to see the trade-offs between the proposed and existing techniques.
  • The references need to be reordered.
  • The idea of pointing out the language errors in the abstract was that the authors would do a thorough language check of the manuscript. E.g., the error on "NoC (Network on Chip) to Network on chip (NoC)" is corrected, but a similar error in the very next line, "MPSoC (Multi-Processor System on Chip)" is not. It would be better if the authors thoroughly checked the language and style of the entire manuscript.
